


# The relationship between Polar Mesospheric Clouds and their background atmosphere as observed by Odin-SMR and Odin-OSIRIS

Ole Martin Christensen[1,2], Susanne Benze[2], Patrick Eriksson[1], Jörg Gumbel[2], Linda Megner[2], and Donal P. Murtagh[1]

[1]Department of Earth and Space Sciences, Chalmers University of Technology, Gothenburg SE 41296, Sweden
[2]Department of Meteorology, Stockholm University, Stockholm SE 11296, Sweden

*Correspondence to:* Ole Martin Christensen
olemartin.christensen@misu.su.se

**Abstract.**

In this study the properties of Polar Mesospheric Clouds (PMCs) and the background atmosphere in which they exist are studied using measurements from two instruments, OSIRIS and SMR, on-board the Odin satellite. The data comes from a set of tomographic measurements conducted by the satellite during 2010 and 2011. The expected ice mass density and cloud frequency for conditions of thermodynamic equilibrium, calculated using the temperature and water vapour as measured by SMR, are compared to the ice mass density and cloud frequency as measured by OSIRIS. Similar to previous studies, we find that assuming thermodynamic equilibrium reproduces the seasonal, latitudinal and vertical variations in ice mass density and cloud frequency, but with a high bias of a factor of 2 in ice mass density.

To explain this bias we use a simple ice particle growth model to estimate the time it would take for the observed clouds to sublimate completely and the time it takes for these clouds to reform. We find a difference in the median sublimation time (2.1 h) and the reformation time (3.2 h) at peak cloud altitudes (82-84 km). This difference implies that temperature variations on these timescales have a tendency to reduce the ice content of the clouds, explaining the high bias of the equilibrium model.

Finally, we detect, and are for the first time able to positively identify, cloud features with horizontal scales of 100 to 300 km extending far below the region of supersaturation (>2 km). Using the growth model, we conclude these features cannot be explained by sedimentation alone, and suggest that these events may be indication of strong vertical transport.



## 1 Introduction

Noctilucent, or Polar mesospheric clouds (PMCs) are clouds that form just below the polar summer mesopause, due to the extremely cold conditions in this region. During the last decades there has been a considerable effort to understand the composition and formation processes of these clouds, and several key features have been discovered. We know that they consist of ice particles (Hervig et al., 2001) with radii around 50 nm (e.g. Thomas and McKay (1985); Baumgarten et al. (2007)). After their formation the ice particles sediment downwards, growing into visible particles while they consume the available water from the ambient atmosphere (e.g. Jensen and Thomas (1988); von Zahn and Berger (2003)).

These clouds are very sensitive to changes in the atmosphere, and as such serve as a useful tool to investigate this otherwise hard-to-reach region of the atmosphere. Observations of PMCs have been used to establish connections between the winter and summer hemispheres (inter-hemispheric coupling) (Becker et al., 2004; Karlsson et al., 2007), as well as between different atmospheric layers (intra-hemispheric coupling) (Karlsson et al., 2009). The question of how long-term changes in the background atmosphere can affect the brightness and frequency of occurrence of these clouds (Thomas et al., 1989; Hervig and Stevens, 2014), and whether any trend in these values has been seen over the last 20 years (e.g. DeLand and Thomas (2015)), is a topic of debate (von Zahn, 2003; Thomas et al., 2003). However, to accurately address these questions a thorough understanding of the cloud micro-physical processes, and how these relate to the background atmosphere in which they occur, is needed (Rapp and Thomas, 2006).

For PMCs to form, the atmospheric conditions must be favourable. Berger and von Zahn (2007) show that cloud nucleation occurs most effectively in regions where the concentration of water vapour exceeds the saturation level by a factor of 10 or more, with the majority of the particles in clouds located at 69°N nucleating about 3 km higher and 9° polewards of the observed clouds. This means that ice particles are transported in the atmosphere though at variety of different background conditions before finally growing into visible clouds. Following trajectories of single ice particles models (Megner, 2011; Kiliani et al., 2013) have shown that cloud growth occurs in bursts, in regions with high supersaturation near the bottom of the clouds, with most of the rapid growth occurring less than 3 hours before observation.

Since cloud growth and sublimation occur over such short time scales and are so dependent on the saturation ratio, measurements of temperature, water vapour concentration and cloud properties should ideally be performed simultaneously. Such studies have been carried out using the Solar occultation For Ice Experiment (SOFIE) on the Aeronomy of Ice in the Mesosphere (AIM) satellite. Employing a 0-D-model assuming thermodynamic equilibrium, Hervig et al. (2009b) conclude that the seasonal ice abundance is controlled mainly by temperature, and that the measured water vapour concentration is a result of cloud formation, rather than a controller of cloud formation. Other such studies include Zasetsky et al. (2009) which used measurements of temperature, water vapour and



PMCs from the Atmospheric Chemistry Experiment - Fourier Transform Spectrometer (ACE-FTS) to determine the equilibrium sizes of the measured ice particles.

Both SOFIE and ACE-FTS perform measurements using solar occultation, this results in measurements at only a few latitudes during an orbit. This means that horizontal structures in the clouds are not resolved by these instruments. Furthermore, due to their limb-sounding geometry, clouds closer to and further away from the satellite than the tangent point will appear to be at lower tangent altitudes than their true altitudes (Hervig et al., 2009a; Eremenko et al., 2005). This means that these instruments must filter out low lying clouds, reducing their ability to investigate the lower edge of the clouds layer.

In this paper we investigate the relationship between polar mesospheric clouds and their immediate surrounding atmosphere using a simple model assuming thermodynamic equilibrium. Furthermore, we determine how far individual cloud pixels are from thermal equilibrium using a growth model similar to Zasetsky et al. (2009). Determining this "time to reach equilibrium" is interesting as it provides information on at what timescales PMCs respond to changes in the background atmosphere, and hence at what timescales a simple thermodynamic equilibrium model can be expected to provide reasonable results. Finally, since these deviations from equilibrium largely exist due to variability in the background atmosphere, quantifying them can provide metrics useful for testing to what degree models capture this variability.

The analysis is performed on a set of measurements performed by the Odin satellite during 2010 and 2011. These measurements were specifically designed to target the summer mesopause region, and allows us to investigate both horizontal and vertical structures in both the clouds and the background atmosphere. The measurements are retrieved using a tomographic approach. This means that the information gained by measuring the same area of the atmosphere from different directions is used in order to better determine inhomogeneities along the line of sight of the instrument. In particular, the tomographic approach allows us to separate low altitude clouds from near and far-field clouds, for the first time providing simultaneous observations of these low lying clouds and their background atmospheric conditions.

## 2 Method

### 2.1 Odin satellite

Odin is a satellite operating in sun synchronous orbit with an inclination of 98° and with an ascending node equator crossing time of 18.00. It was launched in 2001 and carries two instruments on board, the Optical Spectrograph and Infrared Imager System (OSIRIS) and the Sub-Millimetre Radiometer (SMR). The two instruments are near perfectly co-aligned, and as such perform measurements at the same time and place. The main difference is the across-track horizontal field of view of the two instruments. SMR has a resolution of 2.5 km across-track while OSIRIS covers 20 km across track.



During the summer of 2010 and 2011 a special set of measurements focusing only on the regions around PMCs were performed, measuring at tangent altitudes between 75-90 km. Measuring only at these limited tangent altitudes increases the horizontal sampling compared to nominal Odin measurements. This increase in horizontal sampling in turn allows for tomographic retrievals, further increasing the spatial resolution and information content that can be retrieved. The retrievals are performed on each instrument separately before combining the retrieved data.

### 2.1.1 SMR

SMR measures the $H_2O$ emission line at 557 GHz and can retrieve the water vapour concentration and temperature across the entire middle atmosphere (e.g. Urban et al. (2007), Lossow et al. (2009)). The tomographic measurements are used to retrieve water vapour and temperature between 75 and 90 km with a vertical resolution of 2.5 km, a horizontal resolution of 200 km, and with a precision of 0.2 ppmv for water vapour and 2 K for temperature (Christensen et al., 2015). The tomographic SMR measurements are performed using two different instrument configurations called frequency mode 13 and 19. It has been shown that there is a systematic difference in the retrieved water vapour and temperature between these two modes, and following the recommendations of Christensen et al. (2015) we will only use the results from frequency mode 13 in this study, which is the mode most consistent with measurements by AIM-SOFIE and ACE-FTS (within 5 K and 20 % of water vapour volume mixing ratio). This data is available for 15-16 July and 12-13 Aug for 2010, and 16-17 June and 18-19 July for 2011.

### 2.1.2 OSIRIS

OSIRIS tomographic measurements are used to retrieve the local cloud scattering coefficient between 78 and 87 km. The measurements have the possibility to retrieve cloud structures with a horizontal extent of  200 km and a vertical extent of 1 km. The tomographic algorithm used to convert limb-integrated atmospheric line-of-sight radiances into local information of cloud brightness is discussed in detail by Hultgren et al. (2013). Observation of the local scattering coefficient at seven UV wavelengths (277.3 nm, 283.5 nm, 287.8 nm, 291.2 nm, 294.4 nm, 300.2 nm, 304.3 nm) enable the retrieval of ice particle mode radius, number density, and ice mass density. For the retrieval of mode radius assumptions need to be made concerning the particle population. Consistent with earlier studies (Hervig et al., 2009a; Baumgarten et al., 2010; Lumpe et al., 2013) we make the following assumptions:

– Gaussian distribution of particle sizes with a distribution width that varies as 0.39*mode radius but stays fixed at 15.8 nm for mean radii greater than 40 nm.

– Particles are randomly oriented oblate spheroids with an axial ratio of 2.





The PMC microphysical retrieval and resulting uncertainties in cloud brightness and microphysical products are described in detail by Hultgren and Gumbel (2014). To summarise, the ice mass density is retrieved with an accuracy of around $5 \, \mathrm{ng/m^3}$ and the mode radius is retrieved with an accuracy of $10 \, \mathrm{nm}$, these estimates do not include errors resulting from the assumptions in the particle size distribution.

For the first time, the current study shows OSIRIS tomographic results from the northern hemisphere PMC season of 2011. This season was not included by Hultgren et al. (2013) and Hultgren and Gumbel (2014) due to retrieval stability issues for this season, which resulted in unphysical variations of the cloud scattering coefficient with wavelength. While this rendered about half of the orbits recorded during the northern hemisphere 2011 season unusable for this study, a filtering method was developed to ensure the quality of the remaining half of the data. The method consisted in removing each pixel for which the difference in the retrieved scattering coefficient varied unphysically (by visual inspection) across the measured wavelengths. If a large number of pixels are flagged the entire orbit is excluded from the analysis with the result that 36 of the 89 orbits from 2011 are available for use in the current study.

In total, there are 35 tomographic orbits available that provide both SMR data in frequency mode 13 and usable OSIRIS data. Of these, 11 orbits are from July 2010, 12 from August 2010, 4 from June 2011 and 8 from July 2011. Due to the few orbits available from June, only data results from July and August will be shown in this paper when discussing seasonal differences in the data.

## 2.2 The thermodynamic model

In order to quantitatively compare the background atmosphere measured by SMR to the cloud properties measured by OSIRIS, we use a simple model which predicts the expected ice mass density from the observed background atmosphere by simply assuming thermodynamic equilibrium. Similar models have successfully been used previous studies (e.g. Russell et al. (2010) and Rong et al. (2012)) to investigate the relationship between PMCs and the background atmosphere. For the study presented in this paper, the main advantage of such a simple model is that is directly maps the atmosphere observed by SMR into ice mass, without requiring any further assumptions. Furthermore, to estimate the time it would take the observed cloud pixels to reach thermodynamic equilibrium a growth model is used. Both these models are introduced in this section.

### 2.2.1 Growth model

For clouds to form in the atmosphere, the partial pressure of water vapour needs to exceed the saturation vapour pressure. Several expressions have been used to calculate the saturation pressure for water vapour over ice under mesospheric conditions (see e.g. Rapp and Thomas (2006)), and in this study we will use the formula from Murphy and Koop (2005), which is derived from a numerical



integration of the Clausius Clapeyron equation. The saturation pressure is then given by

$$P_{sat} = \exp\left(9.550426 - \frac{5723.265}{T} + 3.53068 \cdot \ln(T) - 0.00728332 \cdot T\right),$$   (1)

where $T$ is the ambient temperature. If the mixing ratio of water vapour is given by $Q_{gas}$ and the mixing ratio of ice (defined as the ratio of water molecules in ice phase to the total number of molecules in the atmosphere) is given by $Q_{ice}$, the total water pressure is

$$P_{tot} = P_{gas} + P_{ice} = (Q_{gas} + Q_{ice}) \cdot p,$$   (2)

where $p$ is the ambient pressure. In this study $T$ and $Q_{gas}$ is taken from the SMR measurements and $Q_{ice}$ from OSIRIS. The pressure is taken from the MSISE-90 model (Hedin, 1991).

In general clouds will grow if the supersaturation ratio $S \equiv \frac{P_{gas}}{P_{sat}}$ is grater than one, and sublimate if the opposite is the case. Hesstvedt (1961) estimated the growth rate of single ice particles as

$$\frac{dr}{dt} = \frac{f}{\rho_{ice}} \sqrt{\frac{m_{h2o}}{2\pi RT}} \cdot (P_{gas} - P_{sat}) \cdot \Phi,$$   (3)

where $r$ is the radius of the ice particle, $\rho_{ice}$ the density of ice, $m_{h2o}$ the molar mass of water, $R$ the molecular gas constant, and $f$ is a sticking parameter which we set to 0.83 following Gadsden (1998). Finally, since the growth/sublimation rate of a particle is proportional to its surface area a factor, $\Phi$, defined as the ratio between the surface area of a non-spherical particle to that of a

volume equivalent sphere, is included (Turco et al., 1982). For spheroids with and axial ratio of 2, $\Phi = 1.095444$. If $P_{gas}$ and $P_{sat}$ is known, Eq. 3 can be numerically integrated for each ice particle to determine $r(t)$. Assuming that the total water content is conserved, the time to reach equilibrium size and the time to sublimate ($r(t) = 0$) can be estimated for single ice particles.

PMCs consist not of a single particle, but an ensemble of particles. However, the exact size dis-

tribution for PMC particles is highly uncertain. Thus, for simplicity, the particles are assumed all to be spheroids with the same radius and with an axial ratio of 2. The radius used is the corresponding mode radius retrieved from OSIRIS, and the number of particles is then determined by ensuring that the ice mass density was equal to the ice mass density retrieved from OSIRIS. This single radius particle size distribution differs from the Gaussian size distribution used in the OSIRIS retrievals.

However, since Eq. 3 is independent of radius, the particle size distribution will not change with time since all particles grow with the same speed. This means that the time it takes for a single radius size distribution to grow to equilibrium is equal to that of a Gaussian. This does not necessarily hold for sublimating clouds, as the smallest particles in the Gaussian distribution sublimates completely. However, once this stage is reached, the total ice remaining cloud parcel is negligible, and thus this

effect will not significantly affect the results presented in this paper.

Furthermore, the growth model presented does not take into account the fact that forming ice on a spherical surface (i.e. small particles) requires more energy than on flat surfaces. This is know as the Kelvin effect and can be accounted for by adjusting the saturation pressure according to:

$$P_{sat}(r) = p_{sat}(\infty) \cdot \exp^{2m\sigma/\rho kTr_n},$$   (4)




where $P_{sat}(\infty)$ is the vapour pressure above a flat surface (i.e. Eq. 1), $m$ the molecular weight of water, $\sigma$ the surface free energy (0.122 J/m$^2$), $\rho$ the density of the ice particle, $k$ the Boltzmann constant and $r_n$ the radius of the nucleation kernel. Since OSIRIS cannot measure particles of the size where this term becomes important, the Kelvin effect will generally not be used in this study. We will only use it in the discussion of reformation of clouds in Sec. 4.

**2.2.2    Equilibrium model**

With enough time, the amount of ice in an air parcel will reach thermodynamic equilibrium, in this case the ice mass density is given by

$$m_{ice} = (P_{tot} - P_{sat})\frac{m_{h2o}}{R \cdot T}. \tag{5}$$

This value will be referred to as the ice mass density in equilibrium or simply "the equilibrium
model". When this level is reached, all the excess water has been converted into ice, and the saturation ratio of the atmosphere is 1. This is rarely the case in the real atmosphere, and assuming thermodynamic equilibrium has been shown to overestimate the ice mass density. Hervig et al. (2009b) found a better agreement between measurements and the equilibrium model by replacing $P_{tot}$ with $P_{gas}$ in equation 5. Rong et al. (2014) found that the overestimation changes depending on the sea-
son, and introduced an additional scaling factor $F$ to further reduce the ice mass, in particular late in the season. In this paper we will not discuss any of these adjustments in depth, but only include results using the model from Hervig et al. (2009b) in Sec. 3.2 to contextualise the results presented.

**3    Results**

**3.1    OSIRIS clouds and background atmosphere**

A first step in comparing the clouds measured by OSIRIS with their immediate background atmosphere is to look at the deviation from the mean background atmosphere in areas where clouds are detected. This is done by identifying all pixels in the OSIRIS measurements where an ice mass density $> 0$ ng/m$^3$ is retrieved. The mean background atmosphere, with and without clouds, at that latitude and altitude is then subtracted from each pixel. This method removes the effect of zonal
differences in cloud and cloud-free pixels, and hence the anomalies in temperature and water vapour are determined. Figure 1 shows mean and standard deviation of these anomalies for cloud and cloud-free pixels. The mean water vapour and temperature profiles across all latitudes are added to these anomaly values to provide context for the comparison.

In general the atmosphere is about 3-4 K cooler when clouds are observed, with a smaller differ-
ence in temperature at lower altitudes. During July a clear depletion of water vapour (1-2 ppmv) can be seen with the strongest effect at the middle of the PMC layer at about 82.5 km, while at higher altitudes and during August the depletion is smaller (1 ppmv). The background water vapour (no



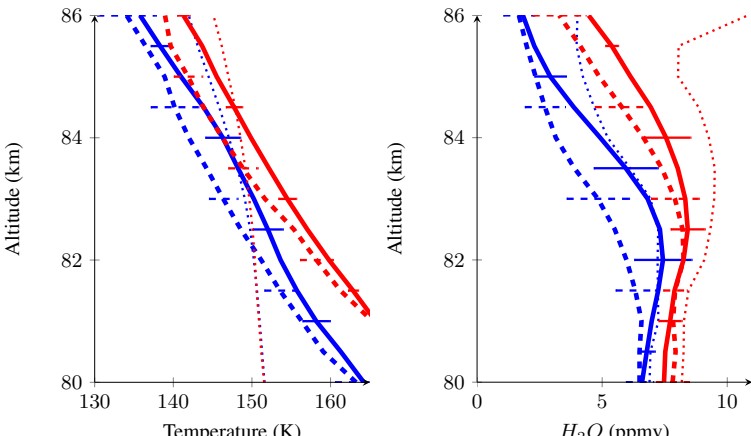

**Figure 1.** The figure shows differences in the measured background temperature and water vapour concentration for cloud (dashed lines) and cloud free pixels (solid lines) for July (blue) and August (red). The values are calculated by adding the measured mean profile for latitudes $> 70\,°$N to the mean anomalies from these latitudes (for further description see the text). The thin horizontal lines are the standard deviation for the plotted quantities. The thin dotted lines show the mean frost point temperature (left figure) and total water content for cloudy pixels ($Q_{tot} = Q_{gas} + Q_{ice}$) (right figure).

clouds) and the total water content for pixels with clouds (ice + vapour) are equal between 80 and 84 km in July, while at higher altitudes, as well as all altitudes in the August data, the pixels with

clouds show an elevated total water content compared to the pixels without clouds. The fact that total water content is preserved in the core cloud altitude region (82-84 km) in July indicates that the growth from sub-visible to visible clouds occurs locally, and thus water vapour is transformed into ice when the conditions allow it. On the contrary, in August and at the highest/lowest altitudes, where enhanced total water content is observed, the water in the clouds is not simply depleted from the lo-

cal surroundings of the clouds, but rather from other areas of the atmosphere. This can for example be due to long formation times at the highest altitudes, or an indication that the clouds/background atmosphere have undergone transport separating the two.

Below PMCs a general enhancement of water vapour is expected, as the clouds sublimate and release available water into the atmosphere. However, Fig. 1 shows no sign of direct water vapour

enhancement under the areas where clouds are detected. This was further investigated by comparing the difference between the mean water vapour concentration and the concentration 0-1.5 km under the clouds, but no measurable enhancement was found. The reason for this is that while for some orbits, enhancements are found under the clouds, this enhancement is highly variable and can exist even when no cloud is observed. Thus, since the deposition of water vapour occurs at the end of the



life cycle of a clouds, there is no direct correlation between individual cloud observations and water vapour enhancements below the cloud.

## 3.2 Water vapour and ice budget

Polar mesospheric clouds are highly sensitive to their background atmosphere, and small errors in the measured temperature or water vapour concentrations used in the calculations described in Sec.

2.2 can give large errors in the equilibrium ice content as well as growth and sublimation times. Thus, before using the data to look at individual measurements, we investigate the ability of the data to reproduce the expected large-scale properties of the ice distribution.

The amount of ice expected in thermodynamic equilibrium can be compared to the ice retrieved from OSIRIS measurements. To take into account the sensitivity of the OSIRIS measurements, the

ice mass density in pixels with ice mass density below a certain threshold is set to 0. The threshold is determined by estimating the average ice mass density needed to reach a cloud brightness of $2 \cdot 10^{-9}$ $\mathrm{m}^{-1}\mathrm{str}^{-1}$ at 83 km. This results in a value of $10.08\,\mathrm{ng/m^3}$ so for simplicity the threshold is set to $10\,\mathrm{ng/m^3}$. The OSIRIS data are also filtered using the same method to ensure that the two datasets are consistent.

### 3.2.1 Vertical Comparison

Figure 2 depicts the overall ice distribution at different altitude. The left figure shows the mean retrieved ice mass density while the right shows cloud presence. In agreement with previous studies we find that assuming thermodynamic equilibrium overestimates the ice mass roughly by a factor of two across the entire region. This indicates that the whole measured region is, on average, highly

supersaturated, with supersaturation ratios greater than 100 often occurring above 84 km. Even in areas where clouds are measured, a considerable supersaturation ($S > 10$) is observed.

Replacing $P_{tot}$ with $P_{gas}$ in Eq. 5, as in Hervig et al. (2009b), reduces the difference between the retrieved ice mass density and the one predicted by the equilibrium model, especially around the peak ice mass density. However, the discrepancy still remains, in particular at the highest altitudes.

Looking at the cloud presence, which we define as the frequency of the number of pixels with an ice mass density above the aforementioned detection threshold, the agreement between the model and the measurements is better than for ice mass density, at least up to 83 km. This indicates that it is the strength of the clouds, rather than the frequency, that leads to the excess mass in the equilibrium model. Below 82 km there are actually more clouds in the OSIRIS data than the equilibrium model

predicts, despite a lower total ice mass density measured. The reason for this is that at the lowest altitudes clouds are detected far outside the saturated region. On average the lower edge of the measured cloud layer is 1 km below the saturated region, with OSIRIS measuring a mean lower edge in July at 82 km and 83 km in August. This will be further discussed in Secs. 3.3 and 3.4.





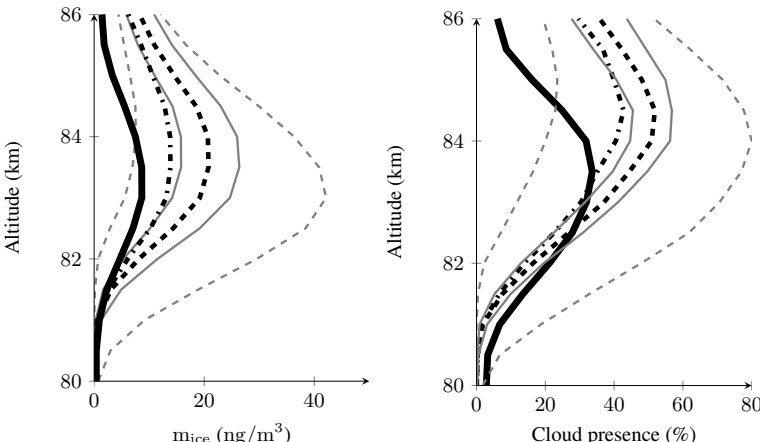

**Figure 2.** Cloud properties in equilibrium with the atmosphere measured by SMR compared to the OSIRIS measurements. The left plot shows the ice mass density inferred from SMR and OSIRIS according to Eq. 5 (in black-dashed) and the result if only $P_{gas}$ is used in Eq. 5 (black-dot-dashed). The black solid line shows the ice mass density measured by OSIRIS. The right plot shows the cloud presence for the same data. The thin gray lines are the results if the calculations using $P_{tot}$ are carried out with a modified atmosphere of $\pm 5\,\mathrm{K}$ (dashed) and $\pm 20\,\%$ water vapour (solid).

In the OSIRIS data a large difference in cloud presence is seen between July and August, with a
presence at the peak of 40% in July and 18% in August. This difference is significantly less in the equilibrium model where the cloud presence in July is 60% and 50% in August. However, in both the measurements and the model the peak presence altitude is about 1 km higher in August than in July.

### 3.2.2  Comparison of Zonal Means

Cloud cover varies with latitude, with higher cloud coverage at higher latitudes than at lower. In Fig. 3 the ice column (integrated ice mass density across all altitudes) and cloud frequency (the percentage of times a cloud pixel with a ice mass density over the given threshold is present at any altitude) is shown for both the OSIRIS measurements and the equilibrium model. The latitudinal variations in the ice column are captured quite well by the model with more ice polewards than at
lower latitudes and during July compared to August. Looking at the cloud frequency however, an interesting discrepancy is visible in the July data: while there are plenty of clouds in the model all the way down to 70 °N the observations show a clear reduction of cloud frequency with latitude. This indicates that at lower latitudes the equilibrium model produces many thin clouds which are not measured by OSIRIS. Possible reasons for this are discussed in Sec. 4.



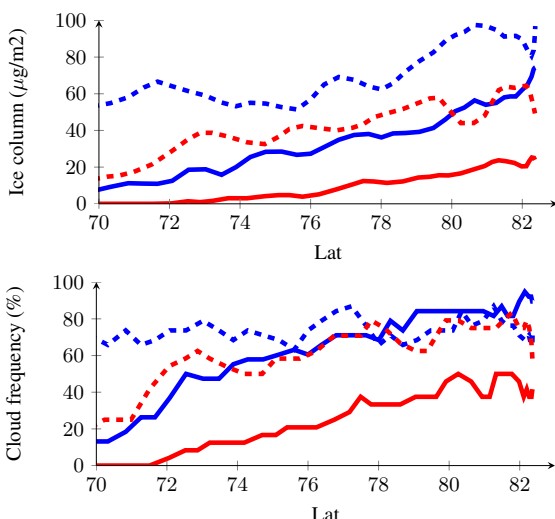

**Figure 3.** The latitudinal distribution of the ice column and the cloud frequency from the equilibrium model (dashed) and OSIRIS (solid) for measurements performed in July (blue) and August (red).

Overall the results from the equilibrium model are in agreement with previous studies. It overestimates ice mass by a factor of ∼2, and the overestimation is larger late in the season than in the middle, in agreement with Rong et al. (2014). The position of the peak ice mass density is reproduced at the same altitude as the measurement and the seasonal variation is seen with a higher ice mass density peak in August compared to July. The latitudinal variation of the integrated water content is also seen in both the equilibrium model and the measurements, while the cloud frequency in

the model shows less latitudinal variation than seen in the measurements, in particular in July.

### 3.2.3   Sensitivity analysis

The ice content predicted by the equilibrium model is very sensitive to changes in temperature and water vapour, and in order to evaluate how any errors in the SMR measurements influence this anal-

ysis, Fig. 2 includes the ice mass density calculated from all SMR measurements with an assumed systematic error of $\pm 5\,\mathrm{K}$ and $\pm 20\%$ water vapour concentration. The results from these perturbed calculations are shown by the thin gray lines. In particular, the strong dependency on temperature is seen as e.g. cloud presence at the cloud peak falls from 60% to 20% if the atmospheric temperature is $5\,\mathrm{K}$ warmer than the temperature retrieved from SMR. This radical change underlines how sensitive

PMCs are to small changes in temperature.

Considering this sensitivity, and the size of the estimated systematic uncertainties in the retrieved water vapour ($> 2$ ppmv) and temperature ($> 5\,\mathrm{K}$) from SMR, the results presented in this study and


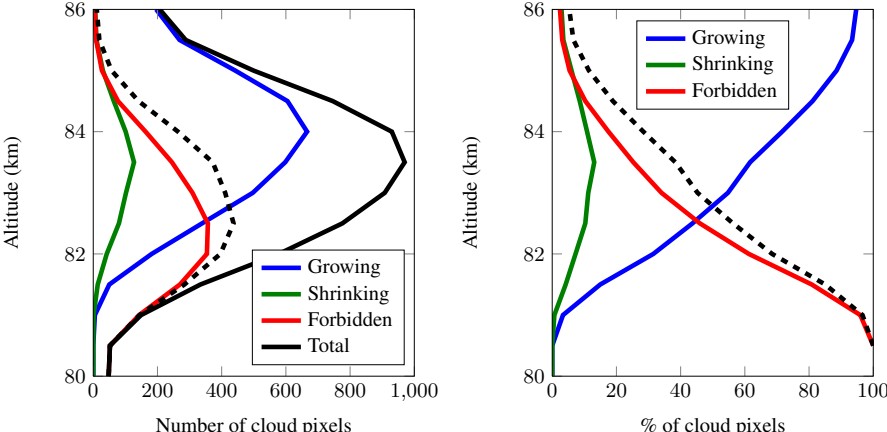

**Figure 4.** Classification of clouds measured by OSIRIS. The left plot shows the number of pixels measured of each type, while the right plot shows the percentage of cloud pixels in each phase. The blue lines show clouds in a growing phase, the green lines indicate that the clouds are sublimating, but that cloud presence is expected in equilibrium, and the red lines show clouds not expected in thermodynamic equilibrium. The black solid line are the total number of clouds detected, while the black dashed lines show the total number of sublimating clouds (i.e. red + green).

previous studies agree remarkably well. In fact, this consistency is a strong indication that the tomographic SMR data is free from substantial biases in temperature and water vapour concentration.

### 3.3 Classification of OSIRIS clouds

As the measurements capture clouds both in growing and sublimating phases, each cloud pixel observed by OSIRIS can be classified based on the state of the background atmosphere. If there is excess water available ($P_{gas} > P_{sat}$), the cloud is growing and thus classified as such. If the amount of ice exceeds that expected from thermodynamic equilibrium, the cloud is classified as shrinking ($P_{gas} < P_{sat}$, but $P_{tot} > P_{sat}$). Finally, if the cloud is outside the region of where any ice should exist in thermodynamic equilibrium ($P_{tot} < P_{sat}$) it is classified as forbidden. For this classification all detected OSIRIS clouds are included (not only those with an estimated ice mass density above the aforementioned threshold).

Figure 4 shows the result from the classification. A majority of the clouds are in a growing phase above 82.5 km, while below this, a majority of the clouds are sublimating, i.e. either classified as shrinking or forbidden. Due to the general downward motion of the ice particles, this altitude, where the number of growing clouds is equal to the number of shrinking clouds, is where the maximum ice mass density is found (see Fig. 2). If only the August data is examined the altitude where more than 50% of the clouds are sublimating is located 1 km higher, at 83.5 km.



Below 82 km almost all the clouds seen are outside the region where clouds should exist if thermodynamic equilibrium is applied. Such "forbidden" clouds have been predicted by models (e.g. Megner (2011), Fig. 7). Feofilov and Petelina (2010) and Hervig et al. (2009b) observe clouds outside the region of saturation. However, since they are unable to distinguish between clouds located in the near and far field of the observed limb, these low altitude clouds are filtered out of the data

as "unphysical". In total about 50 % of the clouds (254 out of 522) observed by Feofilov and Petelina (2010) are filtered out in this process. This number is comparable to our observed ratio where in total 37 % of the observed clouds are outside the region allowed by thermodynamic equilibrium considerations. Unlike Hervig et al. (2009b), we do not see any significant change in the number of these clouds in the August data compared to July. This might be due to the limited data available

in the tomographic dataset. However, it could also depend on the fact that the altitudes where these clouds are found moves upwards with time during a season, and thus fewer of these clouds would be filtered out as near- or far-field clouds in the SOFIE measurements.

### 3.4   Investigation of individual clouds

To look further at how individual measurements of clouds relate to the measured background atmo-

sphere, the equilibrium ice mass density is compared to the measured ice mass density for each orbit separately. Figure 5 shows the measured temperature, water vapour, and calculated ice mass density from the model, together with the measured ice mass density from the OSIRIS data. The three orbits are meant to illustrate some typical features seen across all the orbits measured. The abscissa is given in Angle-Along-Orbit (AAO) which is the effective latitude along the orbit plane, with AAO = 0° at

the equator, and AAO = 90° at the northernmost position of the satellite.

The top panels show an orbit recorded on July 15, 2010. The black curve indicates the area where the supersaturation is greater than 1, and if follows to a large degree 150 K contour line, which previously has been used as a proxy for supersaturation (e.g. López-Puertas et al. (2009)). The clouds measured by OSIRIS (thin contours) are mostly contained within this area. The right panel shows

this in more detail with ice mass density from the equilibrium model as coloured contours and the OSIRIS values as black contours. In general there is a good agreement between the two. However, some of the internal variability seen in the measured clouds is not reproduced by the equilibrium model.

While the top panels show clouds mainly confined within the area of supersaturation, the second

row shows an example where clouds are present outside the region of supersaturation with one strong cloud located at 85° AAO and a thin patch at 98° AAO. These clouds are (assuming that the retrieved water and temperature is correct) undergoing sublimation. Using Eq. 3, the sublimation time for these clouds is estimated to be less than 20 minutes for the strong cloud, and 10 minutes for the thin cloud. Thus the conditions around these clouds must have undergone a rapid change to

allow a detection outside the saturated region.

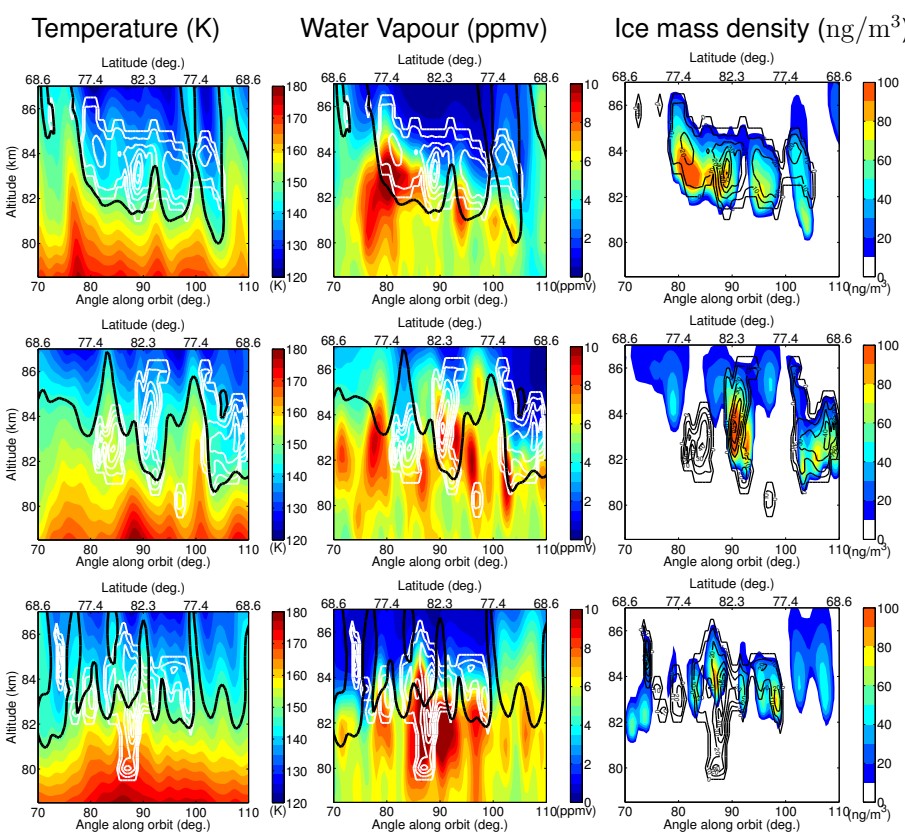

**Figure 5.** Three orbits measured during July 2010 (orbit 51226, top) and 2011 (orbit 56735, mid and orbit 56726, bottom). The left and centre panels show the temperature and water vapour concentration retrieved from SMR (coloured contours) together with the observed ice mass density (white contours, each contour line corresponds to 10 ng/m$^3$). The black contour shows the area where S>1. The rightmost plots show the ice mass density predicted by the equilibrium model (colours) and the ice mass density measured by OSIRIS (black contours, each contour line corresponds to 10 ng/m$^3$).



The lower panels show another case of clouds outside the saturated region, with cloud cover down to 80 km at 87° AAO. These clouds were identified in Hultgren and Gumbel (2014) as regions below the typical cloud brightness peak, with large particles (>80 nm mean radius) "raining out" of the saturated region. To refine this hypothesis, the sublimation time was calculated at different altitudes of the clouds. The resulting sublimation time is around 2 h at 82.5 km, but is rapidly reduced to less than 10 minutes below 81.5 km. Typical sedimentation speeds at 81.5 km for a 100 nm particle is on the order of 0.1 m/s (Turco et al., 1982) thus falling from 82 to 80 km would take more than 5 hours. This indicate that these vertical structures in the clouds cannot come from sedimentation alone.

Horizontal transport (zonal) of the clouds could explain these areas far below the region of supersaturation. However, due to their small sizes, ice particles tend to be transported with the air, hence staying within the same air parcel. There is however a possibility for such vertical features to arise due to wind shear. If there is a strong horizontal gradient in the temperature the supersaturated region can extend to significantly lower altitudes just outside the orbital plane. If the horizontal winds are significantly smaller inside this region of supersaturation than below it, clouds sedimenting out of this region cloud be blown into the orbit plane. The result would be observed as an apparent vertical separation between the $S > 1$ line and the cloud bottom, which would be larger than the true vertical separation. However, investigating orbits preceding and following these events, we do not detect any such strong horizontal gradients in temperature.

Another explanation for these structures is strong vertical winds. Indeed similar vertical cloud features have been measured by LIDARs (Kaifler et al., 2013) and linked to gravity waves propagating through the clouds. Kiliani et al. (2013) also note that ice particles can experience strong downdrafts at the lower edge of the clouds. Using the sublimation time, we estimate that the vertical transport speed needed is on the order of 1-3 m/s, which is larger than what is suggested in Kaifler et al. (2013) and Kiliani et al. (2013) where the vertical transport speeds reported are ~0.1 m/s. Thus, if the observed cloud is due to vertical winds, it is a particularly case with particularly strong downdraft. Such vertical windspeeds (1-5 m/s) have been observed lasting over 1 h in the mesopause region by ground based VHF radar (Rapp and Hoppe, 2006), and thus we believe such winds to be a plausible cause of the observed features.

Finally, it should be mentioned that since the horizontal field of view of the two instruments differ, we cannot rule out that these low lying clouds are caused by dynamical features resolved by SMR but not OSIRIS. But, any such feature would need to be highly localised,

Finally, all three rows in Fig. 5 show areas where the equilibrium model predicts cloud presence, while no cloud is observed by OSIRIS, for example the area between 100° and 110 AAO° in the lowermost panels. Such regions are least common at the highest latitudes and at altitudes between 82-84 km in July, and most common in August, at both low latitudes and high altitudes. Once again this can be explained by temporal variations in the background atmosphere. Since growth time from a sub-visible to a visible cloud is several hours (see Sec. 3.5), the ice particles will not grow into visible





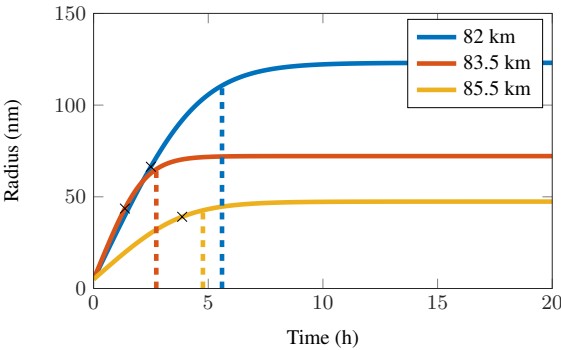

**Figure 6.** Ice particle growth from 5 nm to equilibrium for three cloud pixels at different altitudes observed at AAO = 92° in orbit 56735. The black crosses indicate the measured mode radii from OSIRIS, and the vertical dashed lies show the time when the particle radius has reached 90% of the equilibrium radius.

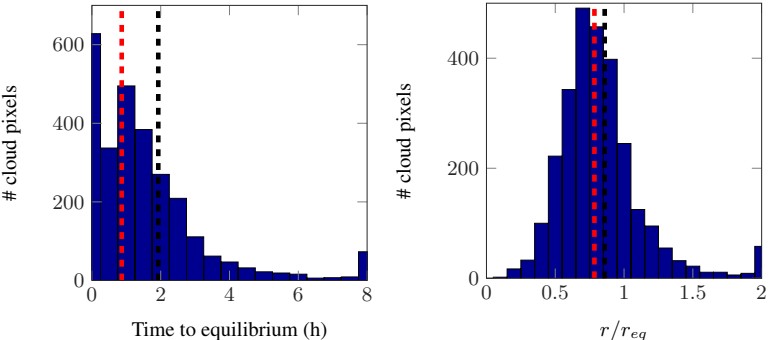

**Figure 7.** Left: Histogram showing the time it will take for the measured pixel to reach equilibrium (not including "forbidden" clouds) for measurements between 82-84 km in July. Right: Histogram showing the ratio between the measured mode radius and the equilibrium mode radius for the same clouds. $r/r_{eq} > 1$ indicates that the cloud is sublimating. The median and mean values are given by the vertical red-dashed and black-dashed lines respectively.

sizes unless these favourable conditions persist. Furthermore, as the largest ice particles sediment downward, only small, and hence sub-visible, particles will remain at the highest altitudes. Both

these effects can lead to areas were cloud presence is predicted by thermodynamic equilibrium, but no cloud is observed. This discrepancy is also reflected in the differences in cloud presence and frequency between the equilibrium model and measurements, shown in Fig. 2 and Fig. 3.



### 3.5 Discussion on the lifetime of clouds

Since the background atmosphere of the clouds is constantly changing, the observed cloud pixels are
always growing or sublimating. To determine to what degree the clouds are in equilibrium (or not)
with their environment, we use the growth model described in Sec. 2.2. For each detected cloud, Eq.
3 is numerically integrated to give an indication of the temporal evolution of the detected cloud. As
the initial condition all ice particles have a radius of 5 nm and $Q_{gas}(t=0) = Q_{tot} - Q_{ice}^{r=5\,\text{nm}}$, where
$Q_{ice}^{r=5\,\text{nm}}$ is the amount of water in ice phase if the cloud consisted of 5 nm particles. This is scaled
such that when the particles have grown to the mode radii measured by OSIRIS, the modelled $Q_{ice}$
is equal to the measured $Q_{ice}$. Figure 6 shows the hypothetical evolution of three cloud pixels from
orbit 56735. The example is taken from the cloud located at $92°$ AAO at three different altitudes. The
figure shows two growth regimes, a fast growth in the beginning due to the large amount of excess
water, and a slower growth as the cloud particles asymptotically grow towards their equilibrium
sizes.

The measured mode radii of the clouds are shown by the black crosses in the figure, and thus give
an indication where along the growth curve each cloud pixel is. From the figure it is clear that the
equilibrium radii decrease with altitude, with equilibrium radii above 100 nm at the lower edge of
the cloud. At the highest altitude, the time it takes for a particle to reach equilibrium is longer than in
the middle of the cloud. This is consistent with the discussion in Sec. 3.1, where no local depletion
of water vapour was seen at the highest altitudes indicating slower cloud growth.

In order to quantify how far a given cloud pixel is from equilibrium, the time from the current
radius to reaching 90% of equilibrium radius is calculated. Figure 7 shows the histograms for all
detected cloud within the core cloud region (82-84 km) in July. It can be seen that a large portion of
the cloud pixels are less than 15 minutes from equilibrium. In total, over 50 % of the clouds are less
than 1 hour from equilibrium, and the mean time to equilibrium is around 2 hours. In terms of radius,
most of the detected cloud pixels are between 0.6-1 times their equilibrium radius, with a mean value
of 0.86. Since the volume of an ice particle scales as $r^3$, while the growth is approximately linear
in time, this rather modest distance from equilibrium (in terms of both time and radius) can lead to
large differences in the observed ice mass density.

While Fig. 7 only considers the pixels where clouds are expected according to thermodynamic
equilibrium, Fig. 8 shows the time to reach equilibrium for pixels where no cloud is expected, i.e.
the sublimation time. The histogram for all these "forbidden" pixels is shown in the left plot in Fig. 8.
It can be seen that a large number of the clouds sublimate fast, and more than 50% are gone after 2.1
hours (dashed vertical line). Thus, as discussed in Sec. 3.4 these clouds indicate areas where a rapid
change of the background atmosphere has occurred, either through transport or temporal changes.

As the clouds seen in a sublimating phase can disappear due to atmospheric variability within
hours, it is of interest to estimate the time it would take to reform these clouds. To estimate this, we
look at the clouds observed in a growing phase, and calculate the time it takes for a 5 nm particle to





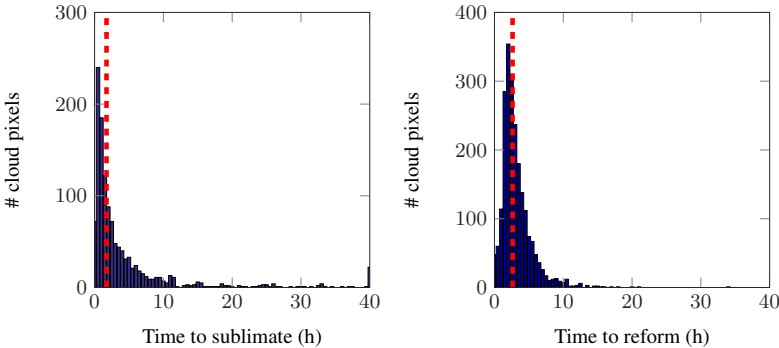

**Figure 8.** Left: The estimated time for "forbidden" clouds between 82-84 km in July to sublimate completely. Right: The estimated time for 5 nm particle to grow to the mode radius observed by OSIRIS. Only growing clouds between 82-84 km in July are considered. The median values are given by the vertical red-dashed lines.

grow to the measured radii. The right plot in Fig. 8 shows the time to reach the measured state for all detected growing clouds, and with a median of 3.2 hours this is significantly longer that the median sublimation time of 2.1 hours in Fig. 8. This asymmetry in cloud destruction and reformation might indeed be one of the reasons why assuming thermodynamic equilibrium overestimates the ice mass density by a factor of two as discussed in Sec. 3.2.

**4  Discussion and conclusions**

In this paper we have compared measurements of cloud ice mass density of polar mesospheric clouds from Odin-OSIRIS with simultaneous measurements of water vapour and temperature from Odin-SMR. The comparison was done on a set of special tomographic measurements performed by Odin, and data from July 2010 and 2011 as well as August 2010 were analysed. We compared the measure-
ments using a model assuming thermodynamic equilibrium. Consistent with previous measurements we find that many general features of the clouds such as the altitude of maximum ice mass density, as well as the latitudinal and seasonal variation of the ice column, are reproduced by the equilibrium model.

Though many large scale features are well represented by the equilibrium model, the model pro-
duces too much ice compared to the OSIRIS observations. This discrepancy has been reported in previous studies. In this study we suggest a possible explanation for this discrepancy. By applying a simple growth model to the clouds observed by OSIRIS we are able to estimate the time to sublimate for the clouds in a shrinking phase, and compare this to the time it would take to grow to the clouds observed in a growing phase. The median time for sublimation in the core cloud region, at 82-84 km
in July, was only 2.1 hours, while the expected time of regrowth was 3.2 hours. This means that



temperature fluctuations on these time scales have a tendency to reduce the total ice compared to an equilibrium situation. The long reformation time also explains why many areas with high amounts of excess water are seen without the presence of PMCs.

One important factor not considered in the discussion above is that although the median reforma-
tion time from 5 nm to the observed radii is 3.2 hours, this reformation time would be longer if the Kelvin effect was included. In fact, rerunning the growth model with the Kelvin effect included we conclude that 20% (433) of the clouds included in the right plot of Fig. 8 would not even be able to reform from 5 nm. This would further increase the depletion of ice expected from short temporal variations in the temperature.

It should be noted that these results do not imply that atmospheric variability in general leads to fewer PMCs. On the contrary, if the mean measured atmosphere is used as an input to the equilibrium model, the amount of ice predicted is significantly less than if the measured atmospheric variability is included. Our results only suggest that atmospheric variability occurring on smaller time scales than $\sim$ 3 hours in the core cloud region (82-84 km in July) has a tendency to reduce the visible ice.
This statement is in agreement with modelling studies. Jensen and Thomas (1994), for example, show that temperature fluctuations with periods of 1 h reduce the observed albedo in the modelled PMCs, and Rapp et al. (2002) conclude that short term variations in temperature reduces visible ice, while variations on time scales larger than 6 hours enhance PMC production.

In the dataset presented we also detect "forbidden" clouds, far below the region of supersaturation
in 5 of the 35 analysed orbits. These have been seen in previous studies using limb sounding tech-
niques, however due to the tomographic nature of the current measurements we are for the first time able to distinguish the low lying clouds from far- and near-field clouds. These cloud regions tend to have small number densities and consist of large particles. The estimated sublimation time for these cloud pixels are on the order of 10-30 min, and since they are found up to 2 km below the region of
supersaturation it is clear that sedimentation alone cannot cause this phenomenon. We suggest that these clouds are linked to strong downdrafts. However, further investigation will be needed to verify this.

Indeed the tomographic measurements by Odin were planned such that they, as much as possi-
ble, would coincide with measurements taken by the Cloud Imager and Particle Size Experiment
(CIPS) on board the AIM satellite. Thus, comparing the results of this study, both in terms of cloud lifetimes and cloud classification, with CIPS images captured a short time before or after the Odin measurement would help refine and verify the conclusions drawn in this paper.

*Acknowledgements.* We would like to thank Mark Hervig for providing the equilibrium model used in this study.



Odin is a Swedish-led satellite project funded jointly by the Swedish National Space Board (SNSB), the
Canadian Space Agency (CSA), the National Technology Agency of Finland (Tekes), the Centre National
d'études Spatiales (CNES) in France and the European Space Agency (ESA).



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
