# Peer review of "The relationship between Polar Mesospheric Clouds and their background atmosphere as observed by Odin-SMR and Odin-OSIRIS"

_Atmospheric Chemistry and Physics, 2016_

## Referee Comment (RC1) · Anonymous Referee #2 · 3 Jul 2016

General comments:

This is a generally well written and interesting study on NLC/PMC observations using a special Odin tomographic measurement mode complemented by model simulations. Measurements with the two Odin instruments SMR (providing T and $H_2O$) and OSIRIS (providing cloud occurrence, ice mass density and particle size information) are used. The dedicated tomographic measurement mode in principle allows distinguishing between clouds in the far/near field – which appear at lower altitudes – and truly low clouds. The modelled ice mass densities are systematically larger than the observed ones, consistent with earlier studies. Possible reasons for this discrepancy are discussed, following similar arguments in earlier studies.

In my opinion the paper is of interest to the aeronomy community and suited for publication in ACP. I ask the authors to consider the specific comments listed below. In addition, it would be good to add brief discussions on (a) how the estimated cloud formation and sublimation times agree with earlier studies, (b) how well the SMR and OSIRIS lines of sight are really aligned – both vertically and horizontally. Fig. 2 shows clear vertical shifts between observed and modelled quantities that may be related to small vertical misalignments of the two instruments. In addition, horizontal misalignments may lead to differences in model and measurement results.

Specific comments:

Line 60: "Both SOFIE and ACE-FTS perform measurements using solar occultation, this results in measurements at only a few latitudes during an orbit"

This is only a minor point, but the statement can be a bit more specific. It's measurements at either 1 or 2 latitudes, depending on whether sunrise and sunset observations are made, or only one type of measurements.

Line 90: "The two instruments are near perfectly co-aligned"

Is it possible to provide a quantitative estimate of the alignment or misalignment of the lines of sight of the two instruments?

Line 117: "Observation .. enable" -> "Observations .. enable" or "Observation .. enables"

Line 166: "is taken" -> "are taken" ?

Line 168: "grater" -> "greater"

Line 176: "is known" -> "are known" ?

Line 188: "sublimates completely" -> "sublimate completely"

Line 189: "the total ice remaining cloud parcel is negligible"

I think something is missing here ?

Line 192: "This is know as" -> "This is known as"

Line 215 – 220: I read this paragraph several times, and I still don't fully understand what specifically was done here. Particularly the sentence "The mean background atmosphere, with and without clouds, at that latitude and altitude is then subtracted from each pixel" irritates me. My understanding is that you determined average T and $H_2O$ profiles for for (a) cloud free and (b) cloudy cases, right? I also don't really understand the statement: "This method removes the effect of zonal differences in cloud and cloud-free pixels .."

I may well be missing an important point, but I suggest rephrasing these sentences.

You also speak of anomalies that are shown in Fig. 1. Usually, an anomaly corresponds to the difference between a given value and its temporal or spatial mean value. This is, however, not the case for the T and $H_2O$ profiles shown in Fig. 1.

Lines 238 – 246: I don't find the argument, why no $H_2O$ enhancement below the cloud is observed, very convincing. This effect has been observed by others (e.g. Hervig et al., JASTP, 2015) and I think there is no reason to assume that the SOFIE observations are wrong. I don't see a problem in simply stating that it is currently not well understood, why an $H_2O$ enhancement below the cloud is not observed. Perhaps the effect would show up in a larger data set?

What about horizontal displacements of the lines of sight between OSIRIS and SMR? The SOFIE T, PMC and $H_2O$ measurements are truly common volume observations. But is this the case for OSIRIS and SMR?

Line 256: ".. to reach a cloud brightness 2 10ˆ-9 / m / str"

Are the units of this 'brightness' correct? It seems like the units are incomplete. It would perhaps be good to clearly state what "brightness" refers to here. The term has different meanings in the literature.
Line 261: "at different altitudeS"

Section 2.3.1 (Vertical comparison): Looking at the two panels of Fig. 2, a vertical shift between the observed and modelled quantities is apparent – slightly more pronounced for cloud presence. It would be good to state the accuracy of the tangent height information of OSIRIS and SMR. Are there any indications for systematic tangent height shifts between the two instruments?

Another question about Fig. 2: the displayed results are averaged over all measurements analyzed?

Line 324: 2A majority .. IS .." and also later in this sentence.

Line 352: "and if follows" -> "and it follows"

Line 414: "is the amount of water in ice phase if the cloud consisted of 5 nm particles"

This means that the 5 nm particles are entirely made up of ice, i.e. a meteoric nucleation nucleus is neglected?

Line 429: "detected cloudS"

Line 446: "longer that" -> "longer than"

---

## Referee Comment (RC2) · Anonymous Referee #1 · 4 Jul 2016

Review of manuscript, "The relationship between Polar Mesospheric Clouds and their background atmosphere as observed by Odin-SMR and Odin-OSIRIS" by O. M. Christen sen et al.

This manuscript contains a detailed analysis of a set of tomographic retrievals of mesospheric ice and accompanying temperature and water vapor measurements. The tomographic nature allows the investigators to determine the three-dimensional character of the clouds, and their relationship to saturation conditions, with 2 km height resolution and 200 km horizontal resolution. They find that many clouds exist in a region of sub-saturation, a result that has eluded previous investigations using line-of-sight observations, which cannot separate out near-field from far-field clouds. This advantage allows them to separate clouds into three classes, depending upon whether the clouds are growing, decaying or should not exist in equilibrium. Among other results, they find tongues of ice particles descending from the cloud base, and ascribe these to large downdrafts, presumably from gravity waves.

This paper is a valuable contribution to the literature, as it breaks new ground in relating mesospheric clouds to their saturation environment. However, I have doubt concerning the equilibrium model's over-prediction of ice from that observed, and the reliance of this to support many of their conclusions. Their agreement with results of previous SOFIE papers that the equilibrium model (or 0D model) predicts a factor of ~2 over that observed is no longer valid, with the release of the new SOFIE version 3 data, which now, because of the lower SOFIE temperatures, yields good agreement of the 0D model with observations. Of course, the authors cannot be held responsible for results not available to them at the time of writing, so this is not a criticism. But if the paper is to be up to date and relevant, they can no longer claim they are in agreement with previously-published SOFIE results. I am not asking that they change their analysis or conclusions, since they clearly rely on their own data, not on SOFIE. However, it appears that the two sets of data are not consistent. It raises the question: if the SOFIE data are closer to reality, and the SMR temperatures are too high, how does this change their conclusions?

I also have a major concern as to why their data do not show water vapor enhancements below the cloud, which are now firmly established as a real effect, occurring at 50 % probability.

Other than these two caveats, I have many small questions and corrections:

Line 37: "*whether any trend..is a subject of debate*". According to Hervig et al (2016), the issue is settled. I recommend that this new reference be cited, and to now please avoid the term "debate" whether or not they agree with the new results and conclusions.

Line 55: "*water is a result cloud formation*'' see Hervig et al. (2015) for an up-to-date study which shows that water can indeed be considered a driver of cloud variability, if the water is averaged over the hydration and dehydration regions.

Line 57: Zasetsky et al (2009) did not only use ACE measurements, they also used a theoretical calculation of ice growth.

Line 188: "*as the smallest particles in the Gaussian distribution sublimates completely*." Should read 'sublimate'

Line 189:" *However, once this stage is reached, the total ice remaining cloud parcel is negligible, and thus this effect will not significantly affect the results presented in this paper*." This sentence needs to be rewritten –awkward with 'effect' and 'affect' in the same sentence.

Line 240-245: "*no sign of direct water vapour enhancement under the areas where clouds are detected.*" The clear detection of this water vapor enhancement (wve) is reported in Hervig et al (2015, JASTP, 132, 124-134) in many solar occultation events. They reported 50% of all observations between May 2007-March 2014 contained wve events. It is my opinion that the authors explanation is weak. Even though they are highly variable, they ought to show up in the averaging! Their sentence "*Thus, since the deposition of water vapour occurs at the end of the life cycle of a cloud(s doesn't belong), there is no direct correlation between individual cloud observations and wve's below the cloud.*" This sentence implies that the very robust SOFIE observations are improbable! Please explain the absence of wve's in the SMR data in a more convincing way!

Line 256: The cloud brightness is given, but at what scattering angle does it apply?

Line 390: "*It is a particularly case with particularly strong winds*" Please restate this sentence. Could 'particularly" be replaced with 'special'?

Line 447: "*This asymmetry in cloud destruction and reformation might indeed be one of the reasons why assuming thermodynamic equilibrium overestimates the ice mass density by a factor of two as discussed in Sec. 3.2.*" Perhaps it is obvious, but I don't understand the reasoning. And it relates to whether the equilibrium model really overestimates the ice mass.

---

## Author Comment (AC1) · 9 Sep 2016

The comment was uploaded in the form of a supplement: http://www.atmos-chem-phys-discuss.net/acp-2016-268/acp-2016-268-AC1-supplement.zip
* * *

---

## Author Response (AR1)

**Reply to referees.**

We would first like to thank the referees for their comments. These will be addressed one at the time in the following reply (referees comments in italic), however we would like to first make some general comments regarding the data used in the paper.

**The OSIRIS dataset**

We are currently in the process of making a more thorough comparison between the tomographic OSRIS dataset and the CIPS instrument on board the AIM satellite. During this process significant differences were discovered between the two instruments at latitudes below 72 degrees due to an error in the background correction of the OSIRIS data. Furthermore, discussions with Nick Lloyd at the University of Saskatoon led to the conclusions that for the data used in the presented study, the reported OSIRIS altitudes are off by a factor of 580 m. The remaining collocation error is less than 100\,m at the tangent point, which is less than the vertical and horizontal resolution of the two instruments.

These two errors have now been corrected, and hence we have decided to update the figures in this paper based on this now corrected dataset. The paragraph describing the OSIRIS data is now rewritten to reflect this.

**Hydration layer below PMCs**

Since both referees commented the fact that we do not find a significant water vapour enhancement under the clouds we wish to address this question first. The two referees comment:

**Referee 1:**
*Lines 238 – 246: I don't find the argument, why no H2O enhancement below the cloud is observed, very convincing. This effect has been observed by others (e.g. Hervig et al., JASTP, 2015) and I think there is no reason to assume that the SOFIE observations are wrong. I don't see a problem in simply stating that it is currently not well understood, why an H2O enhancement below the cloud is not observed. Perhaps the effect would show up in a larger data set?*

**Referee 2:**
*I also have a major concern as to why their data do not show water vapor enhancements below the cloud, which are now firmly established as a real effect, occurring at 50 % probability.*

As the referees correctly state, water vapour enhancements in the lower mesospause is a real and documented effect. The fact that it does not appear in figure 1 has to do with the methodology used to generate the figure and the dataset. Figure 1 describes the relationship between the clouds and their immediate background, i.e. cloud presence is defined on a pixel by pixel case rather than e.g. using the integrated column (as done in Hervig et al., JASTP, 2015).

[Figure]

*Figure 1: Mean difference in water vapour (ppmv) between cloud and cloud-free profiles (IWC > 5 g/km²) for the July OSIRIS data.*

However, as we write later in the paragraph we are also unable to detect any significant increase in water vapour below individual clouds. We want to point out that this does not mean that we do not see enhancement features in the dataset at altitudes corresponding to the bottom of the PMC layer. For example, the three centre panels in figure 5 show several areas with water vapour concentrations exceeded 8 ppmv, indicating a strong hydration compared to what can be considered the unperturbed background atmosphere.

We are however unable to extract any significant correlation between cloud presence and the hydration feature, unlike e.g. Hervig et.al 2015. If we apply a similar methodology as used in Hervig et.al 2015 to our data from July, i.e. classifying profiles as cloud/cloud free if the ice column (or ice water content) exceed 5 g/ km² some indications of an enhancement may be identified (see Figure 1 in this response), however this enhancement is not large enough to be considered significant.

As referee 1 suggests the reason that no significant correlation is found may be due to the limited dataset. In particular, determining the unperturbed background atmosphere is difficult from the tomographic dataset, as only around 20 profiles are available from each measurement period to determine the background for each latitude. Furthermore, recovering the background atmosphere is complicated due to the fact that enhancement features are often seen below both cloud and cloud free areas. Hervig et. al. 2015 used an iterative approach to remove the hydration features for these cloud free profiles. Such an approach requires many profiles at similar latitude and time, and hence is unsuited for the tomographic dataset described in the paper which covers many latitudes over only a few days.

We have changed the paragraph discussing the missing enhancement feature to better reflect the discussion presented above.

**Reply to specific comments by Referee 2:**

*General comments:*

*In my opinion the paper is of interest to the aeronomy community and suited for publication in ACP. I ask the authors to consider the specific comments listed below. In addition, it would be good to add brief discussions on (a) how the estimated cloud formation and sublimation times agree with earlier studies, (b) how well the SMR and*

*OSIRIS lines of sight are really aligned – both vertically and horizontally. Fig. 2 shows clear vertical shifts between observed and modelled quantities that may be related to small vertical misalignments of the two instruments. In addition, horizontal misalignments may lead to differences in model and measurement results.*

With regards to the estimated formation and sublimation times we now compare our results to Kiliani et.al. 2013, which using a Lagrangian cloud growth model coupled to the Leibniz-Institute Middle Atmosphere model find very similar results to us with growth and sublimation times at around 2 hours.

With regards to the misalignment of the two instruments, OSIRIS and SMR are not truly common volume observations, and some differences in the pointing of the instruments are seen as mentioned earlier in this reply. The remaining collocation error is less than 100\,m at the tangent point, which is less than the vertical and horizontal resolution of the two instruments.

Furthermore, although the instruments have a similar resolution in the vertical axis, the resolution of the instruments differ significantly in the horizontal across-track direction, with SMR having a resolution of ~2 km while the resolution of OSIRIS is on the order of 40 km. This means that atmospheric variability on scales between 2-40 km may affect our results in that the atmospheric state retrieved by SMR may represent a special sub-section of the state retrieved by OSIRIS. But, due to the comparatively coarse resolution along-track for both instruments (200 km+) these structures would need to have an almost constant value along the SMR line of sight (i. e. not vary along-track) in order to have any significant effect on the results, and we do not believe such cases are highly unlikely to show up in the dataset.

Note that this does not mean that variability not resolved by the measurements does not influence the data presented, but this is a concern that even applies to common volume measurements like those of SOFIE. Analyzing the effects resolution have on the measured and modelled results requires several assumptions on PMC and atmospheric properties on these smaller scales, and even these assumptions (such as the shape of the particle size distribution of the ice particles) may change depending on scale. Hence, we consider such a discussion far beyond the scope of this paper.

**Specific comments:**

*Line 60: "Both SOFIE and ACE-FTS perform measurements using solar occultation, this results in measurements at only a few latitudes during an orbit"*
*This is only a minor point, but the statement can be a bit more specific. It's measurements at either 1 or 2 latitudes, depending on whether sunrise and sunset observations are made, or only one type of measurements.*

We have now changed the wording to conform with these this suggestion

*Line 90: "The two instruments are near perfectly co-aligned"*
*Is it possible to provide a quantitative estimate of the alignment or misalignment of the lines of sight of the two instruments?*

Based on private correspondence with Nick Lloyd we estimate this to less than 100m

*Line 117: "Observation .. enable" -> "Observations .. enable" or "Observation .. enables"*

This is now corrected

*Line 166: "is taken" -> "are taken" ?*

This is now corrected

*Line 168: "grater" -> "greater"*

This is now corrected

*Line 176: "is known" -> "are known" ?*

This is now corrected

*Line 188: "sublimates completely" -> "sublimate completely"*

This is now corrected

*Line 189: "the total ice remaining cloud parcel is negligible"*
*I think something is missing here ?*

The sentence now reads: "the total ice remaining in the cloud parcel is negligible".

*Line 192: "This is know as" -> "This is known as"*

This is now corrected

*Line 215 – 220: I read this paragraph several times, and I still don't fully understand what specifically was done here. Particularly the sentence "The mean background atmosphere, with and without clouds, at that latitude and altitude is then subtracted from each pixel" irritates me. My understanding is that you determined average T and H2O profiles for for (a) cloud free and (b) cloudy cases, right? I also don't really understand the statement: "This method removes the effect of zonal differences in cloud and cloud-free pixels .."*
*I may well be missing an important point, but I suggest rephrasing these sentences.*
*You also speak of anomalies that are shown in Fig. 1. Usually, an anomaly corresponds to the difference between a given value and its temporal or spatial mean value. This is, however, not the case for the T and H2O profiles shown in Fig. 1.*

We how now rephrased the paragraph so it is hopefully more understandable what has been done in order to create figure 1. The main reason we have to first calculate the zonal mean anomaly before averaging in the meridional direction is that the background atmosphere changes with latitude. If we simply compared the mean atmosphere of cloud-free pixels to cloudy pixels the average of cloudy pixels would be skewed towards higher latitudes, while the cloud-free pixels would be skewed towards lower latitudes. Hence the comparison would be dominated by the meridional differences in the background atmosphere rather than the effect of cloud presence.

*What about horizontal displacements of the lines of sight between OSIRIS and SMR?*
*The SOFIE T, PMC and H2O measurements are truly common volume observations.*
*But is this the case for OSIRIS and SMR?*

See reply earlier in this document

*Line 256: ".. to reach a cloud brightness 2 10ˆ-9 / m / str"*
*Are the units of this 'brightness' correct? It seems like the units are incomplete. It*
*would perhaps be good to clearly state what "brightness" refers to here. The term has*
*different meanings in the literature.*

Brightness is now changed to scattering coefficient which is the more precise term of the retrieved quantity.

*Line 261: "at different altitudeS"*

This is now corrected

*Section 2.3.1 (Vertical comparison): Looking at the two panels of Fig. 2, a vertical shift*
*between the observed and modelled quantities is apparent – slightly more pronounced*
*for cloud presence. It would be good to state the accuracy of the tangent height in-*
*formation of OSIRIS and SMR. Are there any indications for systematic tangent height*
*shifts between the two instruments?*

See response earlier in this reply

*Another question about Fig. 2: the displayed results are averaged over all measure-*
*ments analyzed?*

Yes, this is now explained in the text as well.

*Line 324: 2A majority .. IS .." and also later in this sentence.*
This is now corrected

*Line 352: "and if follows" -> "and it follows"*
This is now corrected

*Line 414: "is the amount of water in ice phase if the cloud consisted of 5 nm particles"*
*This means that the 5 nm particles are entirely made up of ice, i.e. a meteoric nucle-*
*ation nucleus is neglected?*

Yes, this is chosen only to provide a somewhat realistic starting condition.

*Line 429: "detected cloudS"*

This is now corrected

*Line 446: "longer that" -> "longer than"*

This is now corrected

**Reply to Referee 1:**
**General comments:**

*This paper is a valuable contribution to the literature, as it breaks new ground in relating mesospheric clouds to their saturation environment. However, I have doubt concerning the equilibrium model's over-prediction of ice from that observed, and the reliance of this to support many of their conclusions. Their agreement with results of previous SOFIE papers that the equilibrium model (or 0D model) predicts a factor of ~2 over that observed is no longer valid, with the release of the new SOFIE version 3 data, which now, because of the lower SOFIE temperatures, yields good agreement of the 0D model with observations. Of course, the authors cannot be held responsible for results not available to them at the time of writing, so this is not a criticism. But if the paper is to be up to date and relevant, they can no longer claim they are in agreement with previously-published SOFIE results. I am not asking that they change their analysis or conclusions, since they clearly rely on their own data, not on SOFIE. However, it appears that the two sets of data are not consistent. It raises the question: if the SOFIE data are closer to reality, and the SMR temperatures are too high, how does this change their conclusions?*

We thank the reviewer for pointing us to studies using the newer datasets from SOFIE. In particular we notice that there is a better agreement between the 0D model in Hervig 2013 and Hervig 2015 than in the 2009 paper. In these two papers the 0D model overestimates the integrated ice column (IWC) by ~10% for the northern hemisphere (as compared to 35% in Hervig 2009). Only the gas phase water, "Q_gas", is used to estimate the ice mass, though this is based on a cloud-free background profile, and therefore should compensate for the dehydration caused by the presence of PMCs.

If the mean IWC measured by OSIRIS for July is compared to the one produced by the 0D model (using Q_gas) the OSIRIS IWC is about 40% lower. Using non-frequency weighted averages (i.e. excluding entries with zero IWC in the averages) this bias is reduced to 20%. This is, as the referee states, not consistent with the newest SOFIE data.

Comparisons done against the SOFIE V1.2. dataset in Christensen, 2014 shows that, if anything, the SMR dataset is on average colder that the V.1.2 datasets, hence we do not believe that differences seen can be explained by the temperature differences in the two datasets. However, the tomographic PMC data from OSIRIS has not been compared to SOFIE directly, and differences in the sensitivity between these instruments (and possible discrepancies in their retrieved ice mass) could be an explanation for this discrepancy. This would also be consistent with the fact that while we find that the model has a higher cloud frequency than measured, SOFIE finds a lower (Hervig, 2009) or similar (Hervig, 2013) frequency. But even differences in measurement geometry, resolution and latitudinal/temporal sampling could result in different results from the two instruments.

In light of this we have now changed the text to include these considerations, and no longer claim to be in agreement with SOFIE. Furthermore, we have moderated our statement regarding the "explanation of the high bias" statement in the abstract to a "possible explanation" as the reviewer points out that the instrumental data is less consistent than we thought.

**Specific comments:**

*Line 37: "whether any trend..is a subject of debate". According to Hervig et al (2016), the issue is settled. I recommend that this new reference be cited, and to now please avoid the term "debate" whether or not they agree with the new results and conclusions.*

The paragraph is now changed to:
"Furthermore, PMCs are considered to be an indicator of long term changes in the background

atmosphere (Thomas et al., 1989; Hervig and Stevens, 2014), and hence PMC measurements can help establishing trends in temperature and water vapour in the mesopause region where they form (Hervig et al., 2016)."

*Line 55: "water is a result cloud formation" see Hervig et al. (2015) for an up-to-date study which shows that water can indeed be considered a driver of cloud variability, if the water is averaged over the hydration and dehydration regions.*

> The paragraph is now changed to:

Such studies have been carried out using the Solar occultation For Ice Experiment (SOFIE) on the Aeronomy of Ice in the Mesosphere (AIM) satellite. These have shown that many of the critical cloud parameters, in particular cloud-frequency and the integrated ice column (IWC) can be successfully recreated on a seasonal basis by employing a 0-D- model assuming thermodynamic equilibrium Hervig et al. (2009b, 2013).

*Line 57: Zasetsky et al (2009) did not only use ACE measurements, they also used a theoretical calculation of ice growth.*

> Have how now added that a nucleation model is used as well

*Line 188: " as the smallest particles in the Gaussian distribution sublimates completely." Should read 'sublimate'*

> This is now corrected

*Line 189:" However, once this stage is reached, the total ice remaining cloud parcel is negligible, and thus this effect will not significantly affect the results presented in this paper." This sentence needs to be rewritten –awkward with 'effect' and 'affect' in the same sentence.*

> We have now changed "affect" to "impact"

*Line 240-245: "no sign of direct water vapour enhancement under the areas where clouds are detected." The clear detection of this water vapor enhancement (wve) is reported in Hervig et al (2015, JASTP, 132, 124-134) in many solar occultation events. They reported 50% of all observations between May 2007-March 2014 contained wve events. It is my opinion that the authors explanation is weak. Even though they are highly variable, they ought to show up in the averaging! Their sentence "Thus, since the deposition of water vapour occurs at the end of the life cycle of a cloud(s doesn't belong), there is no direct correlation between individual cloud observations and wve's below the cloud." This sentence implies that the very robust SOFIE observations are improbable! Please explain the absence of wve's in the SMR data in a more convincing way!*

> See answer earlier in this document.

*Line 256: The cloud brightness is given, but at what scattering angle does it apply?*

> OSIRIS measures at scattering angles between 70-90 degrees, and the threshold is determined based on the average of all the measured data. To make this clearer the paragraph is now changed to:

"The amount of ice expected in thermodynamic equilibrium can be compared to the ice retrieved from OSIRIS measurements. To take into account the sensitivity of the OSIRIS measurements, the ice mass density in pixels with ice mass density below a certain threshold is set to 0. The scattering coefficient measured depends on the the size and number of ice particles in the cloud, as well as the scattering angle which varies between 70-90 ∘ for OSIRIS. A reasonable threshold was hence determined by estimating the average ice mass density (all clouds, all scatting angles) needed to reach a cloud scattering coefficient of $2 \cdot 10^{-9}$ m$^{-1}$ str$^{-1}$ at 83 km. This results in a value of 10.08 ng/m so for simplicity the threshold is set to 10 ng/m$^3$ . The OSIRIS data are also filtered using the same method to ensure that the two datasets are consistent."

*Line 390: "It is a particularly case with particularly strong winds" Please restate this sentence. Could 'particularly" be replaced with 'special'?*

We have now changed the sentence to read "…, it is a case with a particularly strong downdraft.".

*Line 447: " This asymmetry in cloud destruction and reformation might indeed be one of the reasons why assuming thermodynamic equilibrium overestimates the ice mass density by a factor of two as discussed in Sec. 3.2." Perhaps it is obvious, but I don't understand the reasoning. And it relates to whether the equilibrium model really overestimates the ice mass.*

If the time it takes for a cloud to reform would be much longer than it would take for a cloud to sublimate, a parcel of air which on average should have clouds in equilibrium would with a large probability not have any clouds due to random fluctuations in temperature. Similarly, if the reformation time was much shorter than the sublimation time many parcels of air which on average would not exhibit clouds in equilibrium would have clouds presence reminiscent of some earlier random fluctuation which made it cold enough for clouds to form.  Hence a discrepancy of the sublimation and reformation time of PMCs cloud lead to a high or low bias of the observed clouds (and ice mass) compared to what is predicted in an equilibrium case.

[revised manuscript text omitted]